# FantasyPortrait: Multi-Character Portrait Animation with Expression-Augmented Diffusion Transformers

## Abstract

Producing expressive facial animations from static images is a challenging task. Prior methods relying on explicit geometric priors (e.g., facial landmarks or 3DMM) often suffer from artifacts in cross reenactment and struggle to capture subtle emotions. Furthermore, existing approaches lack support for multi-character animation, as driving features from different individuals frequently interfere with one another, complicating the task. To address these challenges, we propose FantasyPortrait, a diffusion transformer based framework capable of generating high-fidelity and emotion-rich animations for both single- and multi-character scenarios. Our method introduces an expression-augmented learning strategy that utilizes implicit representations to capture identity-agnostic facial dynamics, enhancing the model's ability to render fine-grained emotions. For multi-character control, we design a masked cross-attention mechanism that ensures independent yet coordinated expression generation, effectively preventing feature interference. To advance research in this area, we propose the Multi-Expr dataset and ExprBench, which are specifically designed datasets and benchmarks for training and evaluating multi-character portrait animations. Extensive experiments demonstrate that FantasyPortrait significantly outperforms state-of-the-art methods in both quantitative metrics and qualitative evaluations, excelling particularly in challenging cross reenactment and multi-character contexts.

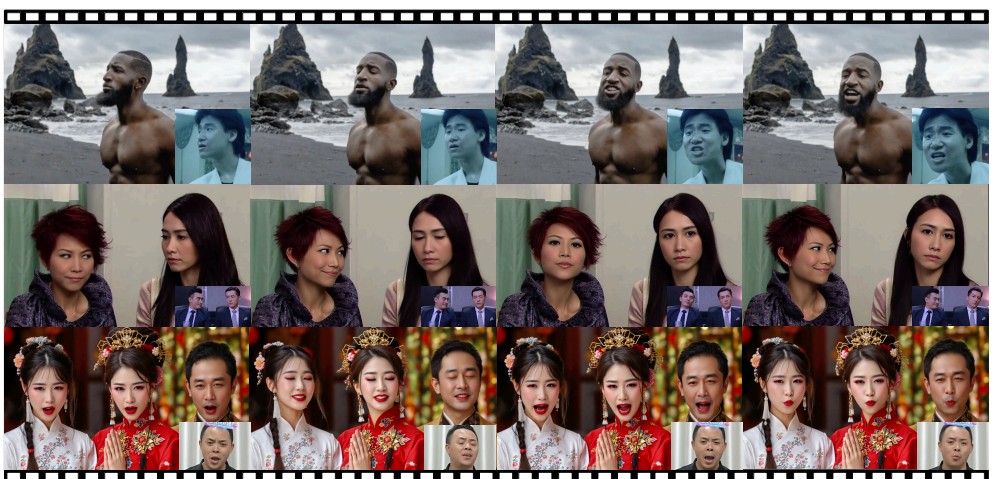

Figure 1: Given a portrait image and a reference motion video, FantasyPortrait generates vivid animated portraits during cross-reenactment. It achieves high-fidelity facial dynamics and natural head movements for both single-character and multi-character.

# 1 INTRODUCTION

Portrait animation aims to generate dynamic facial video sequences from static images, enabling rich and natural expressions with broad applications in film production (Gu et al., 2024), virtual communication (Khmel, 2021), and gaming (Li et al., 2024). Existing approaches typically rely on driving video inputs and employ generative models (e.g., GANs (Zeng et al., 2023; Drobyshev et al., 2022; Wang et al., 2023a; Deng et al., 2024; Guo et al., 2024), NeRF (Yu et al., 2023; Ye et al., 2024), and Diffusion Models (Ma et al., 2024; Xie et al., 2024; Qiu et al., 2025)) to manipulate facial expressions through geometric priors like facial landmarks (Lugaresi et al., 2019) or 3D Morphable Models (3DMM) (Egger et al., 2020).

However, these geometry-based methods face two fundamental limitations. First, they struggle with cross reenactment when significant facial geometry differences exist between the source image and the driving video (e.g., across ethnicities, ages, or genders), often leading to facial artifacts, motion distortions, and background flickering. Second, explicit geometric representations are insufficient to capture subtle expression variations and complex emotional nuances, as they require precise alignment between source and target faces. These issues severely hinder performance in cross-identity scenarios.

Moreover, prior research primarily focuses on single-character portrait animation, shedding little light on multi-character collaborative animation. In such a setting, features from different individuals can interfere with each other, causing expression leakage, where facial attributes of one character inadvertently transfer to others. This makes it challenging to maintain both expression independence and harmony among characters. The absence of publicly available datasets and standardized evaluation benchmarks for multi-character portrait animation further impedes progress in this area.

In this work, we propose FantasyPortrait, a Diffusion Transformer (DiT)-based framework for generating precisely aligned, emotionally expressive multi-character portrait animations. Specifically, we extract implicit expression representations from the driven videos to capture identity-agnostic facial dynamics and enhance the model's ability to express fine-grained affective nuances through expression-augmented learning. To enable coordinated yet independent control of multi-character expressions, we introduce a masked cross-attention mechanism for avoiding inter-character interference. To advance the training and evaluation of multi-character portrait animation, we present ExprBench, a novel benchmark that captures a wide range of expressions, emotions, and head movements across both single- and multi-character settings. Extensive experiments on ExprBench demonstrate that FantasyPortrait consistently outperforms existing methods in both quantitative metrics and qualitative evaluations, particularly in cross-identity reenactment scenarios. In summary, our key contributions are as follows:

- We propose an expression-augmented implicit facial expression control method that enhances subtle expression dynamics and complex emotions through decomposed implicit representations and an expression-aware learning module.
- We design a masked attention mechanism that enables synchronized multi-character animation while maintaining rigorous identity separation, effectively preventing cross-character feature interference.
- We construct ExprBench, a specialized evaluation benchmark for expression-driven animation, along with a multi-character expression Multi-Expr dataset. Extensive experiments demonstrate our method superior performance in both fine-grained controllability and expressive quality.

# 2 RELATED WORK

## 2.1 DIFFUSION-BASED VIDEO GENERATION

Early research on video generation (Chu et al., 2020; Wang et al., 2020; Clark et al., 2019; Balaji et al., 2019)primarily relied on Generative Adversarial Networks (GANs) (Goodfellow et al., 2020). Recently, the groundbreaking progress of diffusion models (Ho et al., 2020) in image generation (Dhariwal & Nichol, 2021; Rombach et al., 2022; Podell et al., 2023) has directly catalyzed a surge of interest in video generation. This field has recently undergone a significant paradigm shift, transitioning from conventional U-Net architectures (Ronneberger et al., 2015) to DiTs (Peebles &

Xie, 2023). U-Net-based approaches (Blattmann et al., 2023; Guo et al., 2023; Wang et al., 2023b) typically extend pre-trained image generation models by incorporating temporal attention layers, thereby equipping them with sequential modeling capabilities for video generation. Although these models have demonstrated remarkable video synthesis performance, the latest DiT architectures (e.g., Wan (Wan et al., 2025) and Hunyuan Video (Kong et al., 2024)) have achieved substantial quality improvements. This is accomplished through the integration of 3D VAEs (Kingma et al., 2013) as encoder-decoders, while combining the sequential modeling advantages of Transformer architectures with advanced techniques such as rectified flows (Esser et al., 2024; Lipman et al., 2022). Moreover, DiT-based models have been successfully applied to diverse scenarios like camera control (Cao et al., 2025; Zheng et al., 2024), identity-preserving (Yuan et al., 2025; Zhang et al., 2025; Liu et al., 2025b), and audio-driven (Wang et al., 2025; Kong et al., 2025; Cui et al., 2025a), demonstrating strong application potential and generalization capabilities.

## 2.2 HUMAN PORTRAIT ANIMATION

Portrait animation generation aims to drive static human portraits into dynamic video sequences by leveraging reference conditions such as video and facial expressions. Early approaches (Guo et al., 2024; Zeng et al., 2023; Drobyshev et al., 2022; Wang et al., 2023a; Gao et al., 2025) primarily employed GANs to learned motion dynamics, while more recent methods based on diffusion models (Xu et al., 2025; Ma et al., 2024; Qiu et al., 2025) have demonstrated significantly stronger generative capabilities. However, most existing approaches rely on explicit intermediate representations as driving signals. These methods exhibit two main limitations. Firstly, due to significant variations in facial features among individuals, methods relying on explicit intermediate representations often struggle to achieve precise alignment when there are substantial differences in facial structure between the reference image and the target portrait, leading to degraded generation quality. Secondly, these methods typically require portrait-specific keypoint adaptation, making them difficult to generalize to multi-character portrait animation scenarios. Moreover, some methods (Xie et al., 2024; Yang et al., 2024a) rely on paired datasets to implicitly model actor-specific appearances or motions, using either paired videos or actor-specific video collections, and they do not support multi-actor reenactment. In this study, we propose a novel DiT-based model architecture that implements implicit feature-driven multiple portrait animation, surpassing previous methods in generation quality and generalization capability.

## 3 METHOD

The overall architecture of FantasyPortrait is illustrated in Figure 2. Given a reference portrait image and a driving video clip containing facial movements, we extract implicit facial expression features from the video sequence and transfer and fuse them into the target portrait to generate the final video output. We propose a novel expression-augmented implicit control method, which is designed to learn fine-grained expression features from implicit facial representations while significantly enhancing the modeling of challenging facial dynamics, particularly in mouth movements and emotional expressions. Furthermore, we propose a multi-portrait Masked Cross-Attention mechanism to achieve precise and coordinated control of facial expressions across multiple characters.

### 3.1 PRELIMINARY

#### 3.1.1 LATENT DIFFUSION MODEL.

Our framework is built upon the Latent Diffusion Model (LDM) (Rombach et al., 2022), which operates in latent space rather than pixel space to enable efficient and stable training. The model employs a pre-trained VAE to establish bidirectional mapping between pixel space and latent space. Specifically, the VAE encoder $E$ transforms input video data $x$ into latent representations $z = E(x)$, and the decoder $D$ reconstructs the latent tokens back into video space. During training, Gaussian noise $\epsilon$ is incrementally added to $z$ through a forward process, producing noised latents $z_t = (1-t)z + t\epsilon$, where $t \in [0, 1]$ is sampled from the logit-normal distribution. Furthermore, we incorporate flow matching (Lipman et al., 2022) to simplify the transformation between complex and tractable probability distributions, facilitating sample generation through learned inverse transformations. The LDM's training objective minimizes the discrepancy between the velocity $v_t$ and the noise predicted by the denoising network $v_\theta$ using the following loss function:

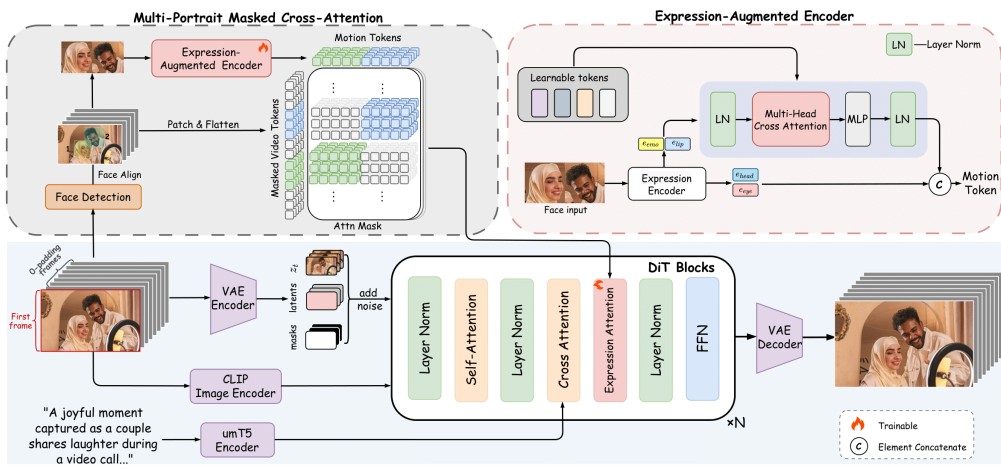

Figure 2: **Overview of FantasyPortrait.**

$$L = \mathbb{E}_{z_t, v_t, t, c} \left[ \| v_\theta(z_t, t, c) - v_t \|_2^2 \right] \tag{1}$$

where $c$ denotes the conditions, $z_1$ denote the latent embedding of the training sample, and $z_0$ represents the initialized noise sampled from the gaussian distribution. The velocity term $v_t = dz_t/dt = z_1 - z_0$ serves as the regression target for the model's prediction task.

### 3.1.2 VIDEO DIFFUSION TRANSFORMER.

Diffusion transformer is an advanced class of diffusion models that employ a multi-layer transformer architecture as the denoising network $u_\theta$, demonstrating exceptional generative capabilities in video synthesis tasks (Seawead et al., 2025; Wan et al., 2025; Kong et al., 2024; Yang et al., 2024b). Specifically, we adopt the Wan (Wan et al., 2025) as the foundational architecture, which consists of 40 transformer layers. The model utilizes a causal 3D VAE to compress videos both temporally and spatially, while incorporating umT5 (Chung et al., 2023) as a multilingual text encoder to effectively integrate textual features via cross-attention mechanisms. Furthermore, Wan enhances conditional generation by integrating a CLIP (Radford et al., 2021) image encoder along with a masked training strategy for initial frames, enabling more effective conditioning on image inputs.

## 3.2 EXPRESSION-AUGMENTED IMPLICIT CONTROL

### 3.2.1 IMPLICIT EXPRESSION REPRESENTATIONS.

We derive identity-agnostic expression representations from the driving video through an implicit feature extraction pipeline. In contrast to conventional portrait animation methods that depend on explicit facial landmarks, our approach leverages implicit encoding to better disentangle expression-related features from confounding factors such as camera motion and subject-specific attributes, thereby achieving more natural and adaptable animation results. Specifically, we detect (Huang et al., 2020) and align the facial region in each frame. Subsequently, we employ a pretrained implicit expression extractor $E_e$ (Wang et al., 2023a) to encode the driving video into expressive latent features. These features include lip motion $e_{lip}$, eye gaze and blink $e_{eye}$, head pose $e_{head}$, and emotional expression $e_{emo}$.

### 3.2.2 EXPRESSION-AUGMENTED LEARNING.

Facial expression generation involves a complex multi-level system, encompassing both relatively simple rigid motion features (e.g., head rotation and eye movements) and highly dynamic non-rigid deformations (e.g., emotion-related muscle activity and lip movements). Simple motions, due to

their more regular patterns and well-defined physical constraints, can be relatively easily modeled. In contrast, complex motions involve richer semantic information and subtle muscle synergies, exhibiting stronger nonlinear characteristics. This significant disparity in feature complexity poses a considerable challenge for simultaneous learning of both motion types.

To address this, we propose an expression-augmented encoder $E_a$ to enhance the learning of subtle and challenging features. Specifically, for $e_{emo}$ and $e_{lip}$, we employ learnable tokens to perform fine-grained decomposition and enhancement, where each sub-feature corresponds to more granular muscle groups or emotional dimensions. Each fine-grained sub-feature then interacts with semantically aligned video tokens via multi-head cross-attention, effectively capturing region-specific semantic relationships. Subsequently, we concatenate the expression-augmented features with $e_{head}$ and $e_{eye}$ to obtain the motion embedding $e_m$, as follows:

$$e_m = Concat(E_a(e_{emo}), E_a(e_{lip}), e_{head}, e_{eye}) \tag{2}$$

### 3.3 MULTI-PORTRAIT ANIMATIONS

#### 3.3.1 MULTI-PORTRAIT EMBEDDINGS.

Using implicit expression-augmented representations, we derive fine-grained portrait motion embeddings for individual characters. For multi-portrait animations, we detect and crop facial regions using the face recognition model (Huang et al., 2020), then extract identity-specific motion embeddings $e_m \in \mathbb{R}^{f \times l \times c}$ for each character, where $f$ is the number of frames, $c$ denotes the number of channel, and $l$ represents the spatial embedding length. The final multi-portrait motion feature $\hat{e}_m$ is obtained by concatenating all $N$ individual embeddings along the length axis:

$$\hat{e}_m = \{e_m^1, e_m^2, \ldots, e_m^N\} \in \mathbb{R}^{f \times (N \times l) \times c} \tag{3}$$

#### 3.3.2 MASKED CROSS-ATTENTION.

To prevent identity confusion and cross-interference between expression-driven signals from different individuals, we design a masked cross-attention mechanism to weight the multi-portrait embeddings in all cross-attention layers. We extract the face mask from the video and then apply trilinear interpolation to map it to the latent space, obtaining the latent mask $M$. The multi-portrait motion embedding $e_m$ interact with each block of the pre-trained DiT through dedicated cross-attention layers. The hidden state $Z_i$ of each DiT block is re-expressed as:

$$Z_i' = Z_i + softmax\left(\frac{M \odot Q_i K_i^\top}{\sqrt{d_K}}\right) V_i \tag{4}$$

where $\odot$ denotes element-wise multiplication, $i$ ndexes the attention block layers, $d_K$ denotes the dimension of keys, $Q_i$ represents the query matrices, $K_i = \hat{e}_m W_i^k$, and $V_i = \hat{e}_m W_i^v$. Here, $W_i^k$ and $W_i^v$ are trainable projection weights for keys and values.

## 4 EXPERIMENT

### 4.1 MULTI-EXPR DATASETS

To address the current scarcity of multi-portrait facial expression video datasets, we introduce Multi-Expr, a novel dataset specifically designed for this purpose. The dataset is curated from OpenVid-1M (Nan et al., 2024) and OpenHumanVid (Li et al., 2025), and we design a comprehensive data processing pipeline—including multi-portrait filtering, quality control, and facial expression selection—to ensure the quality and suitability of the video datasets. First, we employ YOLOv8 (Reis et al., 2023) to detect the number of individuals present in each video clip, and retain only those containing two or more portrait. Next, we filter out low-quality, blurry, or artifact-ridden clips using aesthetic scoring (Yeh et al., 2013) and dathe Laplacian operator. Finally, leveraging facial landmarks detected by MediaPipe (Lugaresi et al., 2019), we compute the angular and motion variations of key facial points to select clips exhibiting clear and expressive facial movements. The dataset comprises approximately 30,000 high-quality video clips, each annotated with descriptive captions generated by CogVLM2 (Hong et al., 2024).

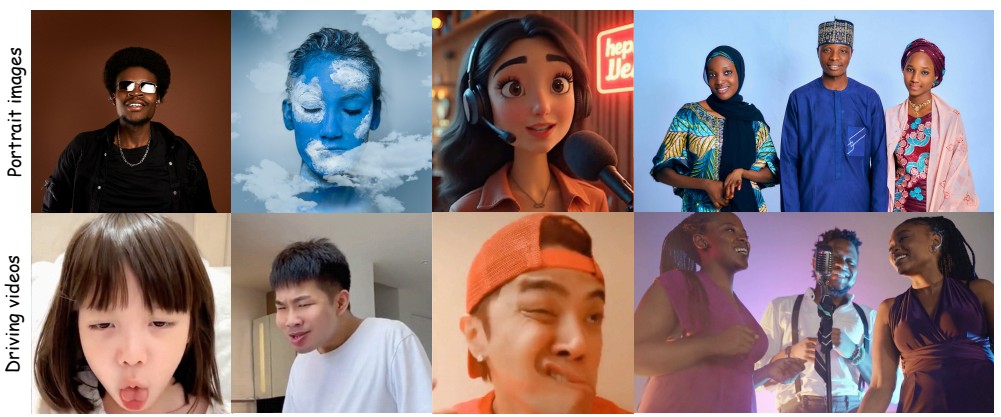

Figure 3: **Examples of ExprBench.**

## 4.2 EXPRBENCH

Due to the lack of publicly available evaluation benchmarks in the field of multiple expression-driven video generation, we introduce ExprBench to objectively compare the performance of different methods in generating facial animations with rich expressions. ExprBench comprises ExprBench-Single for single-portrait evaluation and ExprBench-Multi for multi-portrait scenarios. Specifically, we meticulously collected 200 single portraits and 100 driving videos from copyright-free sources on Pexels (pex) to construct ExprBench-Single. Each driving video was trimmed to 5-second clips containing approximately 125 frames. The portrait images encompass realistic human styles, various anthropomorphic styles (e.g., animals, cartoon characters), and a wide range of scenarios (e.g., recording studios, performance stages, live streaming rooms). The driving videos contain diverse facial expressions (e.g., drooping eyelids, eyebrow twitches), emotions (e.g., happiness, sadness, anger), and head movements.

To further evaluate the performance of multi-portrait expression-driven generation, we also collected 100 portrait images and 50 driving videos to construct ExprBench-Multi, a multi-centric benchmark. ExprBench-Multi is designed to test and compare the performance of different methods in handling video generation tasks involving multiple characters' expressions and movements. Figure 3 showcases examples of portrait images and driving videos from ExprBench.

## 4.3 IMPLEMENTATION DETAILS

We employ Wan2.1-I2V-14B (Wan et al., 2025) as the pretrained model. We train on the single-portrait-centric Hallo3 Dataset (Cui et al., 2025b) and the multi-portrait-centric Multi-Expr Dataset as proposed in Sec. 4.1. The individual embeddings N is 1280. The entire training process runs on 24 A100 GPUs for approximately 3 days, with a learning rate set to 1e-4. To enhance the variability of video generation, we apply independent dropout to the reference image, expression features and prompts, each with a probability of 0.2. We adopt 30 sampling steps during the inference stage. We set the classifier-free guidance scale (Ho & Salimans, 2022) for expressions to 4.5 while keeping text prompts empty.

**Baselines and Metrics.** We select several publicly available portrait animation methods for comparative evaluation in the single-portrait setting, including LivePortrait (Guo et al., 2024), Skyreels-A1 (Qiu et al., 2025), HunyuanPortrait (Xu et al., 2025), X-Portrait (Xie et al., 2024), and FollowYE (Ma et al., 2024). We employ the multiple faces version of LivePortrait as the multi-portrait baseline.

To conduct a comprehensive evaluation, we assess all methods on ExprBench as well as the traditional benchmark dataset HDTF (Zhang et al., 2021). For self-reenactment evaluation, we use the first frame as the source input image and the driving video as ground truth. To assess the generalization quality and motion accuracy of the generated portrait animations, we employ Fréchet Inception Distance (**FID**) (Heusel et al., 2017) and Fréchet Video Distance (**FVD**) (Unterthiner et al., 2019). Additionally, to measure expression motion accuracy, we use Landmark Mean Distance (**LMD**) (Lugaresi et al., 2019), while Mean Angular Error (**MAE**) (Han et al., 2024) is adopted to evaluate eye movement accuracy. For cross-reenactment evaluation, we utilize Average Expression Distance

| Dataset | Method | Self Reenactment | | | | | Cross Reenactment | | |
|---------|--------|-----------|-----------|-----------|-----------|-----------|-----------|-----------|-----------|
| | | FID ↓ | FVD ↓ | SSIM ↑ | LMD ↓ | MAE ↓ | AED ↓ | APD ↓ | MAE ↓ |
| ExprBench-Single | LivePortrait | 79.32 | 438.27 | 0.789 | 6.90 | 9.29 | 45.13 | 39.79 | 17.09 |
| | Skyreels-A1 | 66.84 | 373.98 | 0.812 | 5.21 | 7.59 | 36.91 | 24.27 | 16.41 |
| | HunyuanPortrait | 74.86 | 409.14 | 0.783 | 5.73 | 8.35 | 40.41 | 26.12 | 16.65 |
| | X-Portrait | 83.28 | 445.25 | 0.739 | 7.26 | 9.15 | 49.26 | 29.15 | 18.89 |
| | FollowYE | 103.75 | 489.93 | 0.692 | 9.15 | 12.63 | 54.11 | 32.19 | 21.58 |
| | FantasyPortrait | **64.66** | **358.08** | **0.818** | **5.08** | **6.97** | **33.45** | **23.08** | **14.55** |
| ExprBench-Multi | LivePortrait | 120.43 | 416.39 | 0.737 | 7.36 | 10.57 | 59.09 | 36.14 | 21.52 |
| | FantasyPortrait | **84.09** | **391.12** | **0.765** | **5.34** | **7.42** | **34.63** | **30.64** | **16.26** |
| HDTF | LivePortrait | 65.24 | 402.78 | 0.794 | 6.74 | 9.03 | 46.92 | 37.44 | 16.28 |
| | Skyreels-A1 | 54.55 | 368.93 | 0.823 | 5.16 | 7.38 | 35.48 | 24.93 | 14.23 |
| | HunyuanPortrait | 59.90 | 394.66 | 0.787 | 5.55 | 8.12 | 42.59 | 24.22 | 15.05 |
| | X-Portrait | 71.62 | 427.35 | 0.744 | 7.18 | 8.99 | 49.12 | 31.64 | 16.19 |
| | FollowYE | 101.46 | 453.17 | 0.711 | 8.79 | 11.32 | 52.89 | 30.48 | 20.27 |
| | FantasyPortrait | **56.67** | **338.02** | **0.823** | **4.68** | **6.54** | **30.33** | **29.63** | **13.07** |

Table 1: **Quantitative Results on ExprBench and HDTF.** LMD multiplied by $10^{-3}$, AED multiplied by $10^{-2}$ and APD multiplied by $10^{-3}$. ↑ indicates higher is better. ↓ indicates lower is better.

(**AED**) (Siarohin et al., 2019), Average Pose Distance (**APD**) (Siarohin et al., 2019), and MAE. AED and APD are used to assess the accuracy of expression and head pose movements respectively.

### 4.4 RESULTS.

**Quantitative Results.** The quantitative comparison results are presented in Table 1. The warping-based approach employed by LivePortrait demonstrates limited accuracy in controlling global head movements, resulting in the lowest APD score among the compared methods. Approaches including FollowYE, and Skyreels-A1 utilize explicit facial landmark to control head or facial movements. However, this methodology inevitably introduces identity leakage in cross-reenactment scenarios, consequently degrading performance across AED, APD, and MAE evaluation metrics. Hunyuan-Portrait utilizes implicit signals for facial expression generation, while employing explicit DWPose (Yang et al., 2023) condition to drive head movements, which still exhibits limited performance. The GAN-based method LivePortrait along with UNet-based architectures including HunyuanPortrait, X-Portrait, and FollowYE exhibit inferior FID and FVD scores, indicating their limitations in generated video quality, especially in preserving fine facial details, compared to advanced DiT-based models including Skyreels-A1 and FantasyPortrait. Our method achieves state-of-the-art performance on expression and head movement similarity metrics including LMD, MAE, AED and APD, demonstrating particularly significant improvements in cross-identity reenactment. These results validate that our fine-grained implicit expression representation combined with expression-augmented learning effectively captures nuanced facial expressions and emotional dynamics while maintaining superior cross-identity transfer capabilities. In multi-portrait experiments, our approach also yields the best quantitative results, confirming that the masked cross-attention mechanism enables robust and precise control over multiple portraits.

**Qualitative Results.** Figure 4 presents the qualitative results, demonstrating that our method achieves more accurate facial motion transfer and more visually compelling results. In the single-character case, despite significant interference from camera movement and body pose variations in the driving video, our method still outperforms all baselines in terms of visual quality, while the baselines exhibit artifacts and incorrect expressions under such disturbances. This advantage stems from our expression-enhanced implicit facial control approach, which enables more robust and nuanced expression manipulation. For multi-character scenarios, LivePortrait exhibits noticeable discontinuities between the driven regions and static background areas, as it relies on segmenting and re-compositing the facial regions in the pixel space. In contrast, our method employs a masked cross-attention mechanism that allows for thorough integration of expression features from different identities in the latent space, without mutual interference or leakage between individuals' expressions, thereby producing more natural results.

Figure 4: **Qualitative Results.**

**More Visualization Results.** Our supplementary materials include extended videos showcasing additional visual results, such as diverse portrait styles (e.g., animals and anime characters), outcomes in various complex real-world scenarios (e.g., glasses occlusion, head accessories, and facial obstructions), identity swapping animation, and multi-portrait animation generated by combining multiple single-portrait video inputs.

## 4.5 ABLATION STUDY AND DISCUSSION

| Dataset | Method | AED | APD | MAE |
|---------|--------|-----|-----|-----|
| Single | Ours | 33.45 | 23.08 | **14.55** |
| | Ours(w/o EAL) | 42.88 | 23.10 | 14.57 |
| | Ours(all EAL) | **33.38** | **23.05** | 14.61 |
| | Ours(w/o MCA) | 33.41 | 23.06 | 14.57 |
| | Ours(w/o MED) | 34.02 | 23.15 | 14.63 |
| Multi | Ours | 34.63 | **30.64** | **16.26** |
| | Ours(w/o EAL) | 43.63 | 30.75 | 16.25 |
| | Ours(all EAL) | **34.45** | 30.69 | 16.29 |
| | Ours(w/o MCA) | 73.18 | 46.22 | 24.37 |
| | Ours(w/o MED) | 40.92 | 37.99 | 22.75 |

Table 2: **Ablation Studies in Cross Reenactment.**

| Method | FVD | Speed (s/frame) |
|--------|-----|-----------------|
| Ours | **64.66** | 13.04 |
| Ours(Light) | 67.75 | 0.26 |
| LivePortrait | 79.32 | **0.18** |
| Skyreels-A1 | 66.84 | 7.47 |
| HunyuanPortrait | 74.86 | 4.61 |
| X-Portrait | 83.28 | 3.91 |
| FollowYE | 103.75 | 4.09 |

Table 3: **Inference Acceleration.**

**Ablation on Expression-Augmented Learning (EAL).** To validate the effectiveness of our proposed EAL module, we conducted comprehensive comparisons between three configurations: (1) direct concatenation of all implicit features without EAL (**Ours(w/o EAL)**), (2) applying expression-augmented learning to all implicit features (**Ours(all EAL)**), and (3) our selective approach focusing only on lip $e_{lip}$ and emotional $e_{emo}$ features. As demonstrated in Figure 5 and Table 2, the absence of EAL leads to significantly reduced AED scores, indicating impaired fine-grained expression learning capability. Interestingly, both APD and MAE metrics remain relatively stable across all configurations, suggesting that head pose and eye movements follow more rigid, easily-learned motion patterns, and the benefits of augmented learning are inherently limited for these rigid motions. However, for complex non-rigid motions like lip articulation and emotional dynamics, the performance degradation without EAL becomes pronounced. These findings validate our design rationale for selectively applying emotion augmentation to $e_{lip}$ and $e_{emo}$ features, as full augmentation provides negligible benefits for rigid motions while unnecessarily increasing computational complexity.

**Ablation on Masked Cross-Attention (MCA).** The results in Table 2 and Figure 5 underscore the critical importance of MCA in multi-portrait applications. Without MCA, the facial driving features

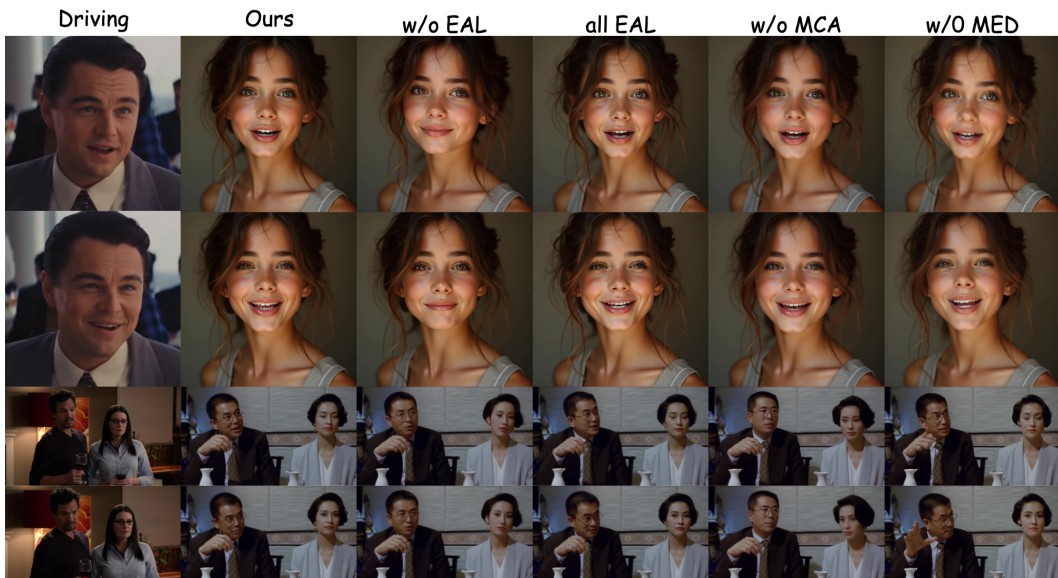

Figure 5: **Qualitative Ablation Results.**

of multiple individuals interfere with each other, leading to significant degradation across all evaluation metrics. As illustrated in Figure 5, the absence of MCA results in mutual interference between characters' facial expressions, generating conflicting outputs that nearly eliminate the model's ability to follow the driving video. In contrast, our designed masked cross-attention mechanism effectively empowers the model to independently control different individuals.

**Ablation on Multi-Expr Dataset (MED).** Our experimental results demonstrate the critical role of multi-expression datasets in portrait animation tasks. As shown in Table 2 and Figure 5, training exclusively on single-portrait datasets maintains comparable performance for single portrait animation, but leads to substantial performance degradation and even visual artifacts in multi-portrait scenarios. These findings demonstrate that while multi-expression datasets may be less essential for single-portrait animation, they are indispensable for achieving high-quality results in complex multi-portrait animation tasks, which facilitates the model's capacity to acquire nuanced facial expression representations across multiple individuals.

**Inference Acceleration.** The iterative sampling process required by diffusion models results in relatively slow generation speeds. To enhance the practicality of FantasyPortrait for broader applications, we developed a light model (Ours(Light)). Specifically, we first replaced the base model with the more cost-effective Wan2.1-I2V-1.3B (Wan et al., 2025). We then employ model distillation (Contributors, 2025), in conjunction with TeaCache (Liu et al., 2025a), to reduce the number of inference steps from 30 to only 4. As shown in Table 3, while the accelerated model makes a slight trade-off in generation quality, it achieves an approximately 50-fold speedup. Compared to previous methods, our accelerated model demonstrates competitive performance in terms of both generation quality and inference speed.

## 5 CONCLUSION

In this work, we present FantasyPortrait, a novel DiT-based framework for generating expressive and well-aligned multi-character portrait animations. Our method leverages implicit facial expression representations to achieve identity-agnostic motion transfer while preserving fine-grained affective details. Additionally, we introduce a masked cross-attention mechanism to enable synchronized yet independent control of multiple characters, effectively soluting expression leakage. To support research in this field, we contribute ExprBench, a comprehensive evaluation benchmark, along with a multi-character facial expression Multi-Expr dataset. Extensive experiments demonstrate that FantasyPortrait outperforms existing methods in both single- and multi-character animation scenarios, particularly in handling cross-identity reenactment and complex emotional expressions.

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
