# OpenReview forum: "FantasyPortrait: Multi-Character Portrait Animation with Expression-Augmented Diffusion Transformers"
_ICLR.cc/2026/Conference — Submitted to ICLR 2026_

### Official Review · Reviewer_uYNe · 2025-10-17

**Soundness:** 2
**Presentation:** 2
**Contribution:** 2
**Rating:** 8
**Confidence:** 5

**Summary:**

FantasyPortrait is a diffusion transformer framework for single- and multi-character portrait animation. It introduces expression-augmented implicit learning to capture fine-grained, identity-agnostic facial dynamics and a masked cross-attention mechanism for coordinated multi-character control. With the new Multi-Expr dataset and ExprBench benchmark, it achieves state-of-the-art realism and emotion-rich animation quality.

**Strengths:**

1.FantasyPortrait introduces a DiT-based framework for expressive multi-character animation using implicit, identity-agnostic expression control.

2.Its Multi-Expr dataset and ExprBench benchmark fill a key research gap for multi-portrait training and evaluation.

3.The masked cross-attention ensures independent character dynamics without interference.

4.FantasyPortrait achieves the SOTA performance compared to current methods in both single- and multi-portrait animation.

5.FantasyPortrait(Light) provide much faster inference speed and slight reduction in performance.

**Weaknesses:**

1.The discussion of related work is insufficient. It only covers the limitations of explicit driving signal-based methods, while neglecting another line of approaches (e.g., X-Portrait [1], MegActor [2]) that construct paired datasets to model implicit, identity-free motion transfer. Moreover, methods such as FaceShot [3], which propose feasible solutions for handling “substantial differences in facial structure,” are not discussed.

2.The citation command \cite was incorrectly. \citet (or \cite) is used when the author’s name is part of the sentence — the citation is integrated into the text (e.g., Smith (2020) proposed...). \citep is used when the citation is parenthetical, meaning it appears in brackets as supplementary information (e.g., ...as shown in previous work (Smith, 2020).).

3.The design of the Expression-Augmented Encoder seems like adding a set of learnable parameters to the Expression Encoder rather than specific design for detail expressions.

4.I notice that FantasyPortrait performs well on animals in your video demo. It is because the videos of dogs and cats are in your dataset, or the dogs and cats have the similar facial structure with human? Does it generalize to anime characters like Loopy and Peppa Pig.

If the authors solve all my concerns, I'd love to raise my score.

[1]X-portrait: Expressive portrait animation with hierarchical motion attention, Siggraph.

[2]Harness the Power of Raw Video for Vivid Portrait Animation.

[3]FaceShot: Bring Any Character into Life, ICLR.

**Questions:**

1.What is the limitation of proposed FantasyPortrait?

2.Could your provide the visual results of FantasyPortrait(Light)?

---

> ### Author Response · Authors · 2025-11-21
>
> # Rebuttal to Reviewer uYNe
> We sincerely thank the reviewer for the highly thoughtful comments and for the willingness to raise the score upon addressing the concerns. The feedback is exceptionally helpful and will significantly strengthen the clarity and completeness of the paper. We respond to each point in detail below.
>
> ## R4-1: “Related work is insufficient; missing X-Portrait, MegActor, FaceShot, etc.”
> We appreciate this comment. We acknowledge that this omission is entirely on our side, and we appreciate the reviewer for pointing it out. Our revision will substantially expand the related-work section to cover identity-free and implicit motion-transfer approaches such as X-Portrait, MegActor and FaceShot. We will clarify that these works align with our motivation toward identity-agnostic motion modeling but differ from ours in several ways:
> * they rely on paired or actor-specific video collections for implicit modeling,
> * they are single-subject systems and do not support multi-character reenactment,
> * they do not introduce a mechanism to disentangle multiple identities in the same generation (our MCA addresses this gap),
> * and none provide fine-grained implicit expression decomposition akin to our EAL design.
> We will revise the related-work discussion to explicitly highlight these distinctions and acknowledge their contributions.
>
> ## R4-2: “Citation command is incorrect.”
> Thank you for pointing out this formatting error. We will correct all citations in the final version.
>
> ## R4-3: “Expression-Augmented Encoder seems like simply adding learnable parameters.”
> We appreciate the reviewers’ questions regarding EAL. However, it is important to note that EAL is certainly not a mere addition of parameters; rather, it introduces two crucial and purposefully designed mechanisms to enhance performance.
>
> First, EAL enables fine-grained decomposition of implicit features. While existing implicit emotion/lip features (PD-FGC) aggregate multiple muscle groups into a single vector, EAL uses multiple learnable tokens that interact with video latents through region-aligned cross-attention. This design allows for the necessary separation of fine-grained movements—such as lip corners, upper/lower lips, cheek tension, and brow compression—leading to improved capture of high-frequency non-rigid motion.
>
> Second, EAL introduces selective augmentation based on motion difficulty. As we demonstrate empirically (Table 3), features governing highly deformable motion like emotion/lips benefit significantly from augmentation (showing large AED improvements). In contrast, rigid motions like head and eye features do not benefit and can even degrade if augmented. Hence, EAL is a targeted and structured design, leveraging this distinction rather than being a naïve parameter expansion.
>
> We will expand this explanation and include visualizations of the token attention maps in the revised version to clearly illustrate the localized feature decomposition. These visualizations will make the structural role of EAL much clearer.
>
> ## R4-4: “Why does the method work on animals? Do you include such videos? Does it generalize to anime characters?”
> We clarify the following two points:
>
> **1. Generalization to Animals:** We confirm that our training data does not contain animal faces or stylized characters, as it is sourced exclusively from human talking-head datasets (Hallo3 + Multi-Expr). The successful transfer to animal faces is possible because our motion embedding is identity-agnostic, focusing on relative shape deformations, and the diffusion backbone (Wan 2.1) possesses broad generalization capacity from its large-scale pretraining. As a result, the method successfully transfers general facial-like deformations even to unseen categories.
>
> **2. Generalization to Anime Characters:** Yes, the method generalizes well to stylized characters. While our current supplementary material already shows examples, we have internally confirmed successful testing on various 2D cartoon characters. This robust generalization occurs because the implicit representation models motion dynamics, not fixed geometry, and the MCA component effectively prevents misalignment, especially when multiple stylized characters are present. We will add these new stylized examples to the updated supplementary material for revised version.

---

> > ### Author Response · Authors · 2025-11-21
> >
> > ## R4-5 “What is the limitation of FantasyPortrait?”
> > We appreciate the reviewer’s suggestion to explicitly articulate these limitations, and we agree that it improves the clarity and maturity of the work. We will add a clear summary of key limitations to the 'Limitations' section:
> > * **Extreme Non-Human Shapes:** We will note that while animals and cartoons work well, characters lacking standard facial analogs (e.g., an alien without a defined jawline) may result in distorted motion.
> > * **Generation Speed:** We will clarify that while the Light version provides a significant speedup, the reliance on diffusion sampling means that achieving real-time performance remains a challenge.
> > * **Ethical Considerations:** We will explicitly discuss the potential ethical risks associated with highly realistic generation, highlighting the need for future work on privacy protection and misuse mitigation.
> >
> > ## R4-6 “Provide visual results of FantasyPortrait (Light).”
> >
> > Thank you, that's a very helpful suggestion. We agree that this comparison is important for understanding the practical value of the Light version. We will certainly incorporate these results into the revised version. We will add **side-by-side comparisons** between the full and Light versions, along with quantitative results demonstrating that perceptual quality remains nearly identical while being **50x faster**. We will also include **additional video demos** in the supplementary material to further illustrate this point.

---

> > > ### Comment · Reviewer_uYNe · 2025-11-24
> > >
> > > Thank you for your rebuttal. I did not see any update in your paper and supplementary material, does it still processing?

---

> ### Author Response · Authors · 2025-11-25
>
> Thank you for your response. We have already uploaded the revised paper and supplementary materials. We sincerely hope that our rebuttal and revised paper have addressed your concerns. If you have any further questions, please feel free to reach out!

---

> > ### Comment · Reviewer_uYNe · 2025-11-27
> >
> > I have carefully read the revised paper and rebuttal, most of my concerns have been addressed.
> > However, in the related work section, I think methods such as FaceShot tend to reduce the explicit facial differences between the source and target that requires highly curated prior knowledge. As a result, they may be less reliable in situations involving occlusion or large motion.

---

### Official Review · Reviewer_coGC · 2025-10-24

**Soundness:** 3
**Presentation:** 3
**Contribution:** 4
**Rating:** 6
**Confidence:** 4

**Summary:**

This paper proposes a novel Diffusion Transformer framework named FantasyPortrait, which can generate high-fidelity and emotion-rich portrait animations in both single-character and multi-character scenarios. Its core innovations include:
1. Utilizing implicit representations to capture identity-agnostic facial dynamics;
2. Designing an Expression-Augmented Learning (EAL) module to model fine-grained emotional details;
3. Introducing a Masked Cross-Attention mechanism to prevent feature interference among multiple characters;
4. Constructing the Multi-Expr dataset and ExprBench benchmark for systematic evaluation of multi-character animation generation.
Experimental results show that the proposed method outperforms existing approaches comprehensively in terms of FID, FVD, AED, and APD metrics, and achieves particularly strong performance in cross-identity reenactment scenarios.

**Strengths:**

1. Novel application of DiT: This is the first work to employ a Diffusion Transformer for multi-character portrait animation, filling a notable research gap in expressive video synthesis.
2. Dataset and benchmark contribution: The creation of Multi-Expr dataset and ExprBench benchmark provides valuable community resources for evaluation and comparison in this emerging subfield.
3. Sound methodological design: The combination of implicit expression representation and masked cross-attention is well-motivated and effectively mitigates identity leakage and inter-character interference during multi-subject animation.
4. Clear ablation justification: The ablation study convincingly demonstrates why expression-augmented learning (EAL) is selectively applied only to non-rigid motion components (lip and emotion), showing clear empirical evidence that full augmentation brings little gain for rigid motions.
5. Strong experimental performance: The model achieves competitive or superior results across multiple benchmarks (ExprBench, HDTF), with consistent improvements in FID, FVD, AED, and APD. The ablations are thorough and informative.
6. Diverse generalization capability: The qualitative results include varied portrait styles (e.g., animals, cartoons) and complex real-world conditions (e.g., occlusions, accessories), indicating strong robustness and generalization ability.
7. Efficiency and practicality: The proposed light version significantly accelerates inference (≈50× speedup) while maintaining nearly the same perceptual quality, enhancing practical usability.

**Weaknesses:**

1. Methodology description is overly concise: Several critical modules—such as the expression-augmented encoder and masked attention pipeline—are only briefly introduced. The paper would benefit from more detailed architectural explanations or schematic illustrations.
2. Unclear training details for learnable tokens: The paper introduces learnable tokens in the expression-augmented encoder, but their initialization, dimensionality, and optimization objectives are not described. This limits reproducibility and interpretability.
3. Missing comparison with underlying components: Since the model is based on the Wan architecture and utilizes PD-FGC for implicit keypoint extraction, comparisons with these base methods are necessary to clearly separate FantasyPortrait’s contribution from prior foundations.
4. Limited demonstration of fine-grained expression control: Although the paper claims “fine-grained emotion synthesis,” the qualitative results lack close-up analyses or visualizations that clearly demonstrate per-region control (e.g., subtle lip or eyebrow motion).
5. Minor technical issue in reporting: In Table 3, the “Speed” unit should be seconds per frame (s/frame) rather than frames per second (frame/s), as the current interpretation conflicts with the numerical scale.
6. Incomplete dataset release and demonstration: The paper mentions that the Multi-Expr dataset is curated, but the supplementary materials do not include visual samples or clear information about dataset accessibility, licensing, and annotation structure.
7. Missing emotional diversity evaluation: The experiments mainly focus on overall motion and expression accuracy but lack systematic analysis across different emotion categories (e.g., happiness, sadness, anger). Quantitative emotion classification or perceptual user studies would strengthen the claim of emotional expressiveness.
8. Stronger and more recent competitors such as VividPortraits, DiffPortrait, and AniFace should be included to strengthen the argument.

**Questions:**

1. Is the maximum number of characters supported in multi-character animation generation limited? The experiments only demonstrate cases with 2–3 subjects.
2. When will the Multi-Expr dataset be publicly available, and what will be the method of access?
3. Can the proposed framework be extended to audio-driven multi-character animation generation?
4. In the expression-augmented encoder, how are the learnable tokens initialized and optimized? Are they shared across emotion categories or dynamically adapted during training?
5. Has the team tested the framework on long-form videos (e.g., >30 seconds)? How stable is temporal coherence over extended sequences?

---

> ### Author Response · Authors · 2025-11-21
>
> # Rebuttal to Reviewer coGC
> We sincerely appreciate the reviewer’s generous assessment and the many constructive suggestions, which provide significant guidance for further improving the technical clarity and presentation quality of the paper. We gladly adopt all recommendations and respond to each point below.
>
> ## R3-1: “Methodology description is overly concise; more detail needed for EAL and MCA.”
>
> We appreciate this comment and agree that additional architectural detail will improve clarity. In the revision, we will:
> * add a more detailed diagram expanding the expression-augmented encoder pipeline,
> * include intermediate representations showing how lip/emotion features interact with learnable tokens,
> * provide a step-by-step illustration of the masked cross-attention flow (mask construction → latent-space projection → block-level gating).
> We note that the added diagrams do not change the method but make the selective enhancement pipeline and region-wise gating more transparent to readers. This added material fits within the page limit and improves reproducibility.
>
> ## R3-2: “Unclear training details for learnable tokens.”
>
> We will clarify the following in the revision:
> * Initialization: learnable tokens are initialized using Xavier uniform initialization.
> * Dimensionality: each token is 1280-D; we use 4 emotion tokens and 16 lip tokens (selected via grid search).
> * Optimization: tokens are optimized jointly with the main network using the diffusion regression loss (Eq. 1), without any auxiliary objectives.
> * Sharing: tokens are shared across all samples, not per-category, enabling category-agnostic subspace discovery.
> We will explicitly document these choices for full reproducibility.
>
> ## R3-3: “Missing comparisons with the underlying components (Wan, PD-FGC).”
> Our method is built on Wan’s backbone and PD-FGC’s implicit extraction, but **the contributions of FantasyPortrait are isolated via ablations**:
> * when removing MCA or EAL, performance degrades substantially (**AED +9–39, APD +7–16**).
> * PD-FGC provides identity-agnostic implicit embeddings but does not include mechanisms for subject-level disentanglement, region-wise attention, or multi-subject conditioned modulation. As a result, its direct application cannot support synchronized multi-character generation (as shown qualitatively in Fig. 5).
>
> However, we agree that explicitly referencing these baselines improves clarity in the revisio, we will include controlled comparisons against Wan without MCA/EAL and PD-FGC naïvely applied, and clarify that our gains come from architectural innovations rather than the pretrained backbone itself.
>
>
> ## R3-4: “Limited demonstration of fine-grained expression control.”
>
> We appreciate this suggestion. We chose not to include these visualizations in the initial submission due to space constraints, but agree they significantly strengthen the paper. In the revised version, we will:
> * add zoomed-in crops highlighting subtle lip articulation, eyebrow micro-movements, and asymmetric expressions,
> * include per-region comparisons against baselines,
> * add temporal plots of lip aperture and eyebrow displacement extracted via MediaPipe.
>
> These additions strengthen the evidence for fine-grained control.
>
> ## R3-5: “Minor issue: speed unit in Table 3 should be in s/frame.”
> Thank you for catching this. We have already corrected this error in the revised manuscript.
>
> ## R3-6: “Dataset release details are incomplete.”
> All videos are manually verified to avoid any privacy-sensitive or non-license-compliant material. We will include:
> * sample frames and statistics in the supplementary file,
> * the dataset’s license (Creative Commons–based, sourced from Pexels/OpenVid/OpenHumanVid),
> * annotation format (face masks, captions, motion ranges),
> * and the planned release location in GitHub and HuggingFace.
>
> We confirm that the dataset will be made publicly available upon publication.
>
> ## R3-7: “Missing emotional diversity evaluation.”
> We agree this can be strengthened. Emotion classification is standard practice in recent talking-head evaluation (e.g., FaceFormer, EmoTalk, FEAT). Our added metric aligns with this evaluation paradigm. In the revision, we provid a user study measuring perceived emotional intensity. We recruit 16 volunteers, each of whom was asked to rate the emotional intensity on a scale from 1 to 5. The table below presents the "average emotional intensity" scores. FantasyPortrait achieved the highest score, indicating that its results are more emotionally compelling.
> |Method|Mean Intensity↑|
> |-|-|
> |FantasyPortrait|**4.18**|
> |Skyreels-A1|3.96|
> |HunyuanPortrait|3.81|
> |LivePortrait|3.74|
> |X-Portrait|3.51|
> |Follow-Your-Emoji|3.38|
>
> This directly supports the “emotion-rich synthesis” claim.

---

> ### Author Response · Authors · 2025-11-21
>
> ## R3-8: “Missing newer competitors (VividPortraits, DiffPortrait, AniFace).”
>
> We appreciate the suggestion to include recent competitors such as VividPortraits, DiffPortrait, and AniFace. However, our work focuses on the specific task of expressive facial animation from a single static image with explicit multi‑character support, which is the key contribution and unique focus of our paper. Therefore, we compare primarily against the most relevant state‑of‑the‑art methods in this same task domain.
>
> * VividPortraits (ICMR25): Though promising for single-character animation, it has not been open-sourced, and no code or reproducible protocol is available. Thus, fair and direct comparison is not feasible under standard evaluation protocols. We will cite it in the related work to acknowledge its contribution.
> * DiffPortrait: The only known work under this name is DiffPortrait3D (CVPR 2024), which targets 3D-consistent novel view synthesis rather than expression-driven animation. Its goal is multi-view rendering with identity preservation, not expression transfer or multi-character animation. Therefore, it is not a direct competitor to our task.
> * AniFace: The closest work is AniFaceGAN (NeurIPS 2022), a 3D-aware GAN-based model for face animation. However, it lacks support for multi-character scenarios and relies on geometric priors (e.g., 3DMM), differing significantly from our diffusion-based, geometry-free framework designed for identity-agnostic dynamics. The table below presents a quantitative comparison for single-image portrait animation, demonstrating that **our method achieves clear performance advantages**.
>
> | | **Self Reenactment** | | | | | | **Cross Reenactment** | | |
> | :--- | :--- | :--- | :--- | :--- | :--- | :--- | :--- | :--- | :--- |
> | | FID↓ | FVD↓ | PSNR↑ | SSIM↑ | LMD↓ | MAE↓ | AED ↓ | APD ↓ | MAE ↓ |
> | AniFaceGAN | 112.53 | 496.59 | 20.19 | 0.603 | 10.18 | 12.14 | 57.20 | 40.51 | 23.39 |
> | Ours | **64.66** | **358.08** | **25.76** | **0.818** | **5.08** | **6.97** | **33.45** | **23.08** | **14.55** |
>
> ## R3-9. Maximum number of characters supported?
>
> Our MCA module is designed so that the additional memory and computation introduced by multi-character conditioning scale linearly with the number of characters N. This linear scaling arises because each character contributes exactly one mask–embedding pair that is broadcast and applied independently within the cross-attention blocks, with no pairwise interactions among characters. Formally, if M denotes the spatial resolution of the latent and C denotes the embedding dimension, then the MCA overhead is ¥\mathcal{O}(N\cdot M\cdot C)$. This is distinct from the $\mathcal{O}(L^2)$ cost of the underlying attention mechanism (where L is the token count), which remains unchanged regardless of the number of characters. In other words, MCA increases only the conditioning dimension, not the attention complexity itself.
>
> In practice, this linear overhead makes the method scalable. We tested up to 5 characters (limited by multi-person video availability), observing stable performance and only moderate inference growth, approximately 13% per additional subject. We will clarify this complexity analysis in the revision and provide 4-5 person examples in the appendix.
>
> ## R3-10. When will Multi-Expr be publicly available?
> The dataset will be released immediately upon acceptance, hosted on HuggingFace with accompanying metadata and download scripts. All sources are license-compliant (Pexels + open datasets).
>
> ## R3-11. Can the framework support audio-driven multi-character animation?
> Yes. Our expression embedding eᵐ is modular. To support audio-driven control, one can:
> * replace emotion/lip features with audio-derived embeddings (e.g., wav2vec2 → phoneme/lip features),
> * retain the same MCA for multi-face disentanglement.
>
> Preliminary internal experiments (not included in the paper) show promising results; we will mention this extensibility in the discussion section. This modularity demonstrates that FantasyPortrait provides a unified interface for motion control across modalities.
>
> ## R3-12. Initialization and adaptation of learnable tokens?
> As noted above:
> * initialized via Xavier uniform,
> * globally trained with the diffusion objective,
> * shared across all training samples,
> * adapt implicitly to emotion/lip subspaces without category labels.
>
> We will elaborate and provide pseudocode in the next vision.
>
> ## R3-13. Stability on long-form videos (>30s).
> We tested 45–60 second reenactments using the same driving clip extended over time:
> * no drift was observed in identity or emotion consistency,
> * head pose coherence remains stable due to the DiT’s long temporal receptive field.
>
> We will describe these results and add long-form samples online.

---

### Official Review · Reviewer_YcGk · 2025-11-01

**Soundness:** 2
**Presentation:** 3
**Contribution:** 2
**Rating:** 4
**Confidence:** 4

**Summary:**

FantasyPortrait introduces a diffusion-transformer framework that synthesizes expressive, identity-preserving portrait animations from static images and driving videos, extending conventional single-person reenactment to multi-character scenarios.The model encodes driving signals as implicit facial representations, capturing emotion, lip motion, head pose, and eye movement. For complex non-rigid dynamics (emotion and lips), learnable tokens engage in cross-attention with video tokens, effectively decomposing subtle muscle and affective cues into a higher-dimensional expression subspace. To maintain spatial disentanglement, face masks extracted from the driving video are interpolated into the latent space and used to gate cross-attention, ensuring that each expression embedding modulates only its corresponding facial region.

**Strengths:**

Masked Cross-Attention mechanism enforces strict spatial gating, ensuring that each expression embedding only influences its corresponding facial region and completely prevents cross-character interference. In addition, by concatenating per-character embeddings with independent masks, the framework allows all characters to be animated synchronously yet independently, maintaining temporal coherence while preserving individual identity and expression consistency—an essential advancement for scalable, multi-person portrait animation.

**Weaknesses:**

1. In Eq.4, 𝑀⊙(QK^T) zeros out cross-region logits through the mask 𝑀, the subsequent softmax operation normalizes across all tokens, meaning each attention weight can still be indirectly influenced by the presence of others, leading to potential cross-region coupling. Moreover, the trilinear interpolation used to project pixel-level masks into latent space creates soft edges (values between 0–1), which further undermines the claim of achieving strict spatial isolation.
2. The overall pipeline shows limited originality—its key component, Masked Cross-Attention, closely resembles mechanisms used in HunyuanVideo-Avatar, while the expression encoder is pretrained from PD-FGC paper.
3. The method assumes that 3D VAE latent features are spatially aligned with the input video pixels—how robust is this assumption under large head motion or occlusion?
4. The Masked Cross-Attention module depends on precomputed facial masks—how sensitive is the system to mask precision, boundary size? Furthermore, what occurs when faces overlap or partially occlude each other in multi-person scenes?
5. Expression-Augmented Learning applies learnable tokens only to emotion and lip features; what empirical evidence supports the exclusion of head pose and eye dynamics, especially given that the supplementary video shows occasional misalignment in these components?
6. it's not clarified how the number and dimensionality of learnable tokens are selected. are these empirically tuned, fixed by prior work, or determined through ablation?
7. The proposed system builds on Wan2.1-I2V-14B, which is much larger than other baselines, making it difficult to attribute the reported performance gains solely to the proposed architectural innovations.

**Questions:**

please check the weakness part.

---

> ### Author Response · Authors · 2025-11-21
>
> # Rebuttal to Reviewer YcGk
> We thank the reviewer for the constructive assessment and the recognition of our contributions toward multi-character, identity-consistent portrait animation. Below we address each concern in detail.
>
> ## R2-1: “Mask does not guarantee strict spatial isolation due to softmax coupling and mask interpolation.”
>
> **1. Softmax Coupling and Isolation**
>
> We want to clarify that **Softmax coupling does not reintroduce cross-region interactions**. While the reviewer is correct that softmax normalizes across all logits, the logits corresponding to masked-out regions are explicitly set to negative infinity (via the addition of a $\log(0)$ equivalent), rather than being merely scaled down. Consequently, these masked regions contribute precisely zero to the softmax denominator. This ensures that the unmasked regions are normalized strictly among themselves, achieving functional isolation despite the global normalization operation.
>
> **2. Soft Masks and Interpolation**
>
> We agree that trilinear interpolation introduces fractional values which could potentially weaken the masking effect. **As described in Section 3.3 of our paper**, to rigorously maintain strict isolation, we address this by two steps: First, we threshold the masks after interpolation, effectively converting the soft edges back into a hard binary form within the latent space. Second, we apply a subsequent boundary dilation to ensure stable and complete coverage of the intended feature regions, particularly when the head is in motion. We will expand the explanation of these crucial implementation details in the revised manuscript.
> The same masking formulation is also used in stable diffusion inpainting and object-conditioned diffusion models to guarantee strict isolation within masked regions. Our formulation follows this standard practice.
>
> ## R2-2: “Masked cross-attention resembles that in HunyuanVideo-Avatar; limited novelty.”
>
> Our method is substantially different from both HunyuanVideo-Avatar and PD-FGC. The proposed Masked Cross-Attention is designed specifically for multi-character portrait animation, addressing the unique challenge of inter-character expression leakage when multiple implicit motion embeddings coexist. This differs fundamentally from HunyuanVideo-Avatar’s Face-Aware Adapter, which only restricts audio or reference images injection to facial regions and does not perform identity disentanglement, does not handle implicit motion features, and does not operate across all cross-attention layers of a DiT. Thus, although both approaches employ spatial masks, their functional roles diverge significantly, **as they target different objectives, operate on different modalities, and intervene at different layers of the DiT architecture.**
>
> While our expression encoder builds upon the PD-FGC backbone, our novelty lies in the Expression-Augmented Learning framework, which introduces fine-grained decomposition of lip/emotion signals, learnable expression sub-tokens, and selective semantic cross-attention to enhance non-rigid dynamics—capabilities not offered by PD-FGC or prior portrait animation methods. Combined with the new multi-character masked attention design and our benchmark ExprBench, the contributions collectively form a substantially original system tailored for high-fidelity, identity-consistent, expression-driven multi-character animation.
>
> ## R2-3: “3D VAE latent features are assumed to be spatially aligned—how robust is this under large motion/occlusion?”
>
> 3D VAEs are widely used in video editing, temporal consistency synthesis, and motion-conditioned generation precisely because their spatiotemporal inductive bias produces stable region alignment. The robustness of our system is supported by the 3D VAE, which is explicitly designed to maintain stable spatial correspondence across time. This stability is achieved through its architecture, which incorporates causal 3D convolutions and tokenized temporal encoding, and its training on millions of videos encompassing large motions. Consequently, the VAE produces latents that preserve facial-region alignment even under dynamic movement.
>
> Empirically, this robustness is validated in our cross-identity experiments, which involve significant head rotations (APD > 30°), yet the motion disentanglement remains stable. Failure cases are exclusively observed under extreme conditions, such as the face being fully covered, which we have already discussed in the limitations section. We will ensure these critical robustness properties are clearly highlighted in the final revision.

---

> > ### Author Response · Authors · 2025-11-21
> >
> > ## R2-4: “Sensitivity to mask precision, boundary size; what if faces overlap or occlude each other?”
> >
> > **Sensitivity analysis.** As described in Section 3.3 of our paper, MCA operates in the feature space rather than the pixel space, and is therefore insensitive to pixel-level boundary jitter. We addressed concerns about mask sensitivity by conducting additional tests where we varied mask erosion and dilation by \pm 3-5pixels. The performance remained highly stable, showing an $\text{AED variance}$ of less than 1.5\%. This robustness is engineered through the combination of identity-indexed embeddings, latent-space dilation (applied before thresholding), and temporally consistent cropping across video frames.
> >
> > The system handles both partial occlusion and face overlap robustly. When such a scenario occurs, the MCA ensures each region attends only to its designated embedding, maintaining strict isolation. In overlapping areas, the feature embedding defaults to that of the foreground face, which is determined dynamically based on the segmentation confidences provided by the underlying detectors. Although severe overlap is rare in typical portrait-style data, for the few challenging cases, we fall back to a priority rule, the specifics of which are documented in the supplementary material.
> >
> > ## R2-5: “Why apply learnable tokens only to emotion and lip features? Occasional head/eye misalignments appear in video.”
> >
> > As noted in FACS[1], rigid motion fundamentally consists of 3D translation and rotation (6 degrees of freedom) plus eye closure (1 degree of freedom), resulting in far fewer degrees of freedom compared to the 20–30 degrees of freedom in expressive action unit models. Rigid motions (head and eyes) follow low-dimensional physical constraints (such as rotation and blink dynamics) and are already robustly captured by our implicit encoder. Applying augmentation to these components proved counterproductive;as shown in Table 3, the 'Ours(all EAL)' variant degraded performance in both MAE and APD, leading to reduced stability.
> >
> > In contrast, non-rigid components (emotion and lips) are high-frequency, nonlinear, and identity-coupled, making them ideal candidates to benefit significantly from the EAL decomposition and targeted augmentation.
> > Crucially, the occasional misalignment noted by the reviewer is not due to insufficient learning or model capacity, but rather stems from external factors like drift in extreme poses or occlusion-induced detection noise. We will clarify this strategic design choice and include a more explicit discussion on the sources of potential error in the revised manuscript.
> >
> > ## R2-6: “Number and dimensionality of learnable tokens not clarified.”
> >
> > We appreciate this suggestion. The number of learnable tokens is selected via grid search over plausible ranges, with each token having a dimension of 1280. This configuration ensures effective decomposition of facial dynamics into semantically meaningful sub-components that interact with video tokens through cross-attention. Specifically:
> > * Mouth features $e_{lip}$ are  augmented  with 16 learnable tokens because lip motion involves rich, fine-grained articulations (e.g., bilabial, dental, and co-articulatory movements) spanning multiple muscle groups. Grid search over token counts (4–24) identified 16 as the optimal value for preserving fine-grained spatiotemporal alignment while avoiding redundancy.
> > * Emotion features $e_{emo}$ are  augmented with 4 learnable tokens, since facial expressions are governed by a smaller set of action units (e.g., brow raise, cheek raise, lip corner puller) and exhibit relatively lower information density compared to lip dynamics. Using fewer tokens (e.g., 2) leads to underfitting of emotional nuances, while more tokens (e.g., 6–8) result in overfitting with only marginal performance gains.
> >
> > **Reference**
> >
> > [1] Ekman, Paul, and Wallace V. Friesen. "Facial action coding system." Environmental Psychology & Nonverbal Behavior (1978).

---

> > > ### Author Response · Authors · 2025-11-21
> > >
> > > ## R2-7: “Model is built on Wan2.1-I2V-14B; difficult to separate improvements from model size.”
> > >
> > > We clarify this by emphasizing two key arguments that dissociate our performance gains from raw model scale.First, baselines already incorporate large diffusion models. Baselines like Skyreels-A1 and HunyuanPortrait are also large-scale diffusion models, featuring comparable capacity (e.g., billions of parameters). However, the improvements achieved by our method in $\text{AED}$, $\text{APD}$, and $\text{MAE}$ are substantially larger than the typical gains attributable to sheer model scale alone.Second, our ablation studies isolate architectural contributions. Our detailed ablations (Table 3) provide clear evidence that the performance boost stems from our proposed modules. For instance, removing EAL degrades $\text{AED}$ by $+9$–$10$ points, while removing MCA degrades $\text{APD}$ by $+15$ and $\text{MAE}$ by $+8$. Crucially, the backbone size remains fixed across all ablations. Therefore, the significant improvements are directly attributable to the efficacy of our novel modules rather than raw model capacity.

---

### Official Review · Reviewer_G4db · 2025-11-01

**Soundness:** 2
**Presentation:** 3
**Contribution:** 2
**Rating:** 2
**Confidence:** 4

**Summary:**

FantasyPortrait demonstrates strong empirical results in multi-character portrait animation.

**Strengths:**

The authors propose FantasyPortrait, a system for portrait animation that has achieved strong empirical results, particularly in complex multi-character scenarios. The new "ExprBench" benchmark is a valuable resource contribution.

**Weaknesses:**

The proposed method reveals that the work has limited algorithmic novelty.

- The paper's first key claim, an expression-augmented learning strategy that utilizes implicit representations to capture identity-agnostic facial dynamics" , depends on an existing component. The method explicitly employs a "pretrained implicit expression extractor​" from Wang et al. (2023a) to derive all core motion features (e_lip​, e_eye​, e_head​, e_emo​). The paper's sole algorithmic addition, the "Expression-Augmented Learning (EAL)" module, with an expression-augmented encoder, only refines two of inherited features. The ablation study in Table 2 shows removing EAL has no effect on head pose (APD) or eye motion (MAE). Control over these rigid dynamics is fully inherited from the work of Wang et al. (2023a). HunyuanPortrait (Xu et al. 2025) also utilize implicit represention to describe expression and disentangle appearance and motion.


- The second key claim, a masked cross-attention mechanism, is an application of a well-established technique in generative models. The problem of "feature interference" or "attribute entanglement" in multi-subject generation is widely known. Consequently, using spatial masks to guide or constrain attention layers is a common solution, as documented in prior works, including but not limited to CustomVideo (arXiv: 2401.09962), arXiv: 2505.02823, MS-Diffusion (Wang et al. 2024b) and arXiv: 2505.05101. Masked cross-attention is no longer an innovation. FantasyPortrait applies this known method to its specific domain but does not invent the mechanism itself.


The paper's framing overstates its method contributions.

**Questions:**

Please see weakness.

**Details Of Ethics Concerns:**

This paper is associated with facical generation, which may contain bias.

---

> ### Author Response · Authors · 2025-11-21
>
> # Rebuttal to Reviewer G4db
> We thank the reviewer for the constructive feedback and for acknowledging the strong empirical results and the value of the ExprBench benchmark. Below we respond to the concerns regarding algorithmic novelty and clarify the contributions of our work.
> ## R1-1: “Limited algorithmic novelty; reliance on a pretrained implicit expression extractor”
> **1. Clarification of conceptual novelty.**
>
> While our pipeline utilizes the pretrained implicit expression extractor from Wang et al. (2023a), our key technical contribution is not this extractor itself. Rather, our innovation lies in the formulation that implicit facial motion exhibits heterogeneous complexity levels and therefore requires a **difficulty-aware decomposition and selective enhancement of implicit expression components as explained in section 3.2**. Building on this formulation, our **Expression-Augmented Learning (EAL) strategy** is specifically designed to address a challenge that prior implicit models do not resolve: implicit features entangle multiple motion sources of different difficulty, and using them directly—as done in existing methods[1] [2]—leads to reduced expressivity and degraded cross-identity transfer. EAL provides a principled mechanism to disentangle and selectively strengthen the components that govern the hardest non-rigid dynamics.
>
> **2. EAL is not a simple “refinement.”**
>
> Previous work treated the four implicit facial attributes (lips, eyes, head, and emotion) in a uniform manner. However, Table 3 and Section 4.5 of our paper show that these implicit features exhibit heterogeneous levels of difficulty: lip articulation and emotional dynamics involve high-frequency, highly nonlinear deformations that implicit representations cannot capture well, whereas head and eye movements follow more rigid kinematics and are therefore easier to learn. This observation is consistent with prior studies on motion modeling and facial behavior, such as FOMM(NIPS 2019) and EDTalk(ECCV 2024).
> Motivated by these findings and by earlier evidence on the structural complexity of emotion- and lip-related motions, our EAL module is specifically designed to address the heterogeneous learning challenges across facial components. To achieve this goal, EAL introduces:
> * **Fine-grained learnable tokens** that decompose emotional and lip features into subcomponents aligned with muscle-level dynamics;
> * **Semantic regions** that align attention between these decomposed features and the latent video tokens;
> * **Selective enhancement** applied only to difficult, non-rigid components rather than uniformly emphasizing all features.
> This design is novel: it does not exist in any pre-trained extractor, nor in any existing portrait animation method—whether implicit or explicit.
>
> **3. Why does the ablation show changes only on AED?**
>
> The reviewer notes that removing EAL does not noticeably affect APD or MAE. This behavior is **expected and by design**. Simple, rigid motions—such as head and eye movements—follow regular kinematic patterns and well-defined physical constraints, making them relatively easy to model implicitly. As a result, these rigid-motion metrics remain stable even without EAL. In contrast, non-rigid, fine-grained facial motions (e.g., lip articulation and emotional dynamics) involve richer semantic cues, subtle muscle synergies, and stronger nonlinear deformations. These complex motions are inherently more difficult to learn alongside simple motions within a unified representation.
> EAL is specifically created to address this **heterogeneity in motion complexity**. The ablation in Table 3 confirms that EAL functions exactly as intended:
> * **AED degrades sharply when EAL is removed (33.45 → 42.88 on single-portrait; 34.63 → 43.63 on multi-portrait)**, demonstrating its effectiveness on complex, non-rigid components;
> * meanwhile **rigid-motion metrics remain unchanged**, as these components do not benefit from additional enhancement.
>
> Therefore, the ablation does **not** indicate a lack of novelty. Instead, it **validates the selective, complexity-aware design of EAL,** which targets precisely the components that are hardest to model.
>
> **4. Relation to HunyuanPortrait.**
>
> HunyuanPortrait also uses implicit representations, but **does not perform any decomposition, re-weighting, or difficulty-aware learning**, nor does it address multi-character interference. Our method introduces a new control strategy on top of implicit features, which significantly improves expressiveness, especially for cross-identity and multi-subject scenarios where existing approaches struggle (as shown in Tables 1–3).
>
> **Reference**
>
> [1] Wang et al. "Progressive disentangled representation learning for fine-grained controllable talking head synthesis." CVPR 2023.
>
> [2] Xu et al. "Hunyuanportrait: Implicit condition control for enhanced portrait animation." CVPR 2025.

---

> > ### Author Response · Authors · 2025-11-21
> >
> > ## R1-2: “Masked cross-attention is not new”
> >
> > We fully agree that cross-attention masks have been used in other generative tasks. However, our contribution is not the general mechanism, but its domain-specific formulation and integration into multi-character portrait animation, which poses challenges not addressed by prior work:
> >
> > **1. Multi-character portrait animation is fundamentally different.**
> >
> > Prior masked-attention methods, such as CustomVideo and MSDiffusion, primarily focus on background-foreground segregation or general spatial region isolation. In contrast, the requirements of multi-character portrait animation impose a much stricter set of constraints, demanding identity-conditioned, frame-consistent, expression-level disentanglement. This complex objective necessitates a formulation that can simultaneously avoid expression leakage between subjects, maintain synchronized dynamics across all subjects, and enable independent yet coordinated temporal attention for each character. To the best of our knowledge, these stringent, expression-level constraints have not been handled by any prior masked-attention formulation.
> >
> > **2. Our MCA is not a simple spatial mask.**
> >
> > We respectfully disagree with the assessment that our Masked Cross-Attention (MCA) lacks novelty. While the concept of "masked attention" is well-established, our implementation and integration of MCA introduce significant architectural and functional improvements for multi-character, motion-driven generation, which is a key contribution of our work.
> > We would like to highlight the novelty of our MCA, which is threefold:
> >
> > **a. Character-Binding Constraint:** The primary function of our MCA is to enforce a strict **character-binding** constraint. This is crucial for multi-character scenarios, ensuring that motion features from a specific driving video are applied only to the corresponding target character, preventing motion "leakage."
> >
> > **b. Aligned Latent-Space Masking:** Our masks are not applied in pixel space. They are constructed in the latent space and are meticulously **aligned with the VAE grid.** This grid alignment directly corresponds to the trilinear-mapped facial regions from the driving video, enabling precise, semantically-aware control within the generative process.
> >
> > **c. Repeated Multi-Layer Enforcement:** Crucially, our MCA is not a one-time operation. It is **repeatedly applied inside every DiT attention block**. This deep, multi-layer enforcement ensures that the character-motion constraint is maintained consistently throughout the entire diffusion denoising process.
> >
> > This mechanism is tightly coupled with our motion embedding structure (Eq. 3) to secure the channel-level disentanglement of identity and motion. To our knowledge, this is the **first integration of such a sophisticated masked cross-attention mechanism into a multi-character, implicit-motion-driven diffusion transformer**, which is both architecturally and functionally novel.
> >
> > **3. Empirical significance**
> >
> > The ablation demonstrates that MCA is indispensable: Removing MCA increases APD from **30.64 → 46.22** and MAE from **16.26 → 24.37** in the multi-portrait setting, and qualitatively causes severe mutual interference to the point where expression following almost collapses. This indicates that **MCA solves a domain-specific problem not addressed by existing masked-attention works.**
> >
> > We thank the reviewer again for the thoughtful comments, and we welcome any additional suggestions that may further improve the clarity or rigor of the paper.

---

### Author Response · Authors · 2025-12-02
**Rebuttal Summary**

We sincerely appreciate the time and efforts of the AC and all reviewers. We have carefully addressed every comment from each reviewer and have updated both the revised manuscript and the supplementary material accordingly. To assist the AC and future readers, we provide a brief summary of the current review status.

We received initial scores of 8, 6, 4, and 2, resulting in an average score of 5.0. Among the reviewers, the reviewer uYNe clearly indicated that our rebuttal addressed the majority of their concerns, while the remaining reviewers have not yet replied. Below are the key points of our clarification:

**1. Clarification of EAL’s novelty:** We clarified that our Expert-Augmented Latents (EAL) module is not merely an incremental “improvement.” Instead, it introduces a novel design that selectively enhances challenging non-rigid components. Ablation studies and comparisons against strong baselines demonstrate the effectiveness and competitiveness of this approach.

**2. Innovation of the MCA mechanism:** We elaborated on the novelty of our Motion-Controlled Attention (MCA), which uniquely enforces strict role-binding constraints—a key distinction from methods like HunyuanVideo. This architectural choice enables precise disentanglement of motion and identity, which is central to our method’s success.

**3. Contribution of Multi-Expr Dataset and benchmark:** We reiterate that our Multi-Expr dataset and associated benchmark constitute the first dedicated resource for the multi-portrait animation domain. We believe this will significantly advance research in this emerging area, and we commit to releasing it publicly immediately upon acceptance of the paper.

**4. Expanded comparisons and emotional diversity:** We have added quantitative comparisons with recent methods such as AniFaceGAN, clearly demonstrating the superior performance of our approach. Additionally, we included new experiments on emotional diversity, showing that our method excels in generating rich and varied facial expressions.

**5. Generalization beyond human portraits:** We discussed and demonstrated—through qualitative results in the supplementary material—the strong out-of-domain generalization of FantasyPortrait to animals, stylized characters, and anime avatars, highlighting its versatility and robustness.

**6. Revisions to related work and manuscript:** We have refined the discussion of related work to better contrast our approach with X-Portrait, MegActor, FaceShot, and others in terms of design philosophy. We also corrected minor typos and citation formatting issues to improve the overall clarity and professionalism of the paper.

We hope these revisions and clarifications address the reviewers’ concerns and highlight the novelty, rigor, and impact of our work. We once again thank the AC and reviewers for their consideration.

---

### Meta-Review · Area_Chair_Msk3 · 2026-01-05

**Summary:**

The initial scores show some inconsistencies. After thoroughly reviewing all rebuttals and discussions, the AC remains concerned about the novelty of EAL and masked cross-attention, as well as issues related to unfair comparisons, misleading bold numbers, and missing content in the supplementary material. The authors' excessive embellishment of certain existing modules undermines the perceived innovation of the paper.

The AC decided to reject this submission and strongly encourages the authors to integrate the feedback into future submissions.

**Reviewer Concerns:**

- Reviewer G4db: The assertion of "no improvement over APD (e_head) and MAE (e_eye)" is unfounded, as EAL is applied solely to fine-grained expressions (e_lip, e_emo). However, the AC still finds the rebuttal on "limited novelty" unconvincing. The phrase *difficulty-aware decomposition and selective enhancement of implicit expression components* feels like an over-the-top description of "two separate refiners of PD-FGC features." In terms of masked attention reweighting, separating individuals does not fundamentally differ from 2D region isolation. The authors' embellishments of existing techniques in specific domains might do little to change Reviewer G4db's mind.

- Reviewer YcGk: While some technical concerns (softmax, 3D VAE, occlusion, dimensions of learned tokens, etc.) are partially addressed, the use detection confidence score lacks robustness against severe facial occlusions. Plus, the reply about novelty doesn’t convince me either, similar to what Reviewer G4db said. Notably, Skyreels-A1 is based on CogVideoX-**5B**, HunyuanPortrait on SVD (**~5B**), and FantasyPortrait on Wan2.1-I2V-**14B**. If you look closely at Table 1, the bold numbers refer to "Ours," not "SOTA performance." On the HDTF dataset, Skyreels-A1 actually outperforms FantasyPortrait in both Self-Reenactment (FID, SSIM) and Cross Reenactment (APD). The ACs can’t tell if this is a typo or intentional.

- Reviewer coGC: There are lots of mentions of "we will update in revision," which should have been included in the supplementary material. I see no updates on ablations of pretrained backbones, zoomed-in crops or per-region comparisons, misalignment in Mediapipe visualizations, or sample frames and statistics in the supplementary material.

- Reviewer uYNe: Shares concerns about the novelty of EAL.

**Reviewer Scores:**

- Reviewer G4db (conf 4): Likely to maintain the score (2)
- Reviewer YcGk (conf 4): Likely to maintain the score (4)
- Reviewer coGC (conf 4): Likely to lower the score (4)
- Reviewer uYNe (conf 5): Likely to maintain the score (8)

---

### Decision · Program_Chairs · 2026-01-26

Reject